# Synthesis of Carbon Nanomaterials from Biomass Utilizing Ionic Liquids for Potential Application in Solar Energy Conversion and Storage

**DOI:** 10.3390/ma13183945

**Published:** 2020-09-07

**Authors:** Kudzai Mugadza, Annegret Stark, Patrick G. Ndungu, Vincent O. Nyamori

**Affiliations:** 1School of Chemistry and Physics, University of KwaZulu-Natal, Private Bag X54001, Durban 4000, South Africa; mugadzakudzie@gmail.com; 2SMRI/NRF SARChI Research Chair in Sugarcane Biorefining, School of Engineering, University of KwaZulu-Natal, Durban 4041, South Africa; 3Energy, Sensors and Multifunctional Nanomaterials Research Group, Department of Chemical Sciences, University of Johannesburg, Doornfontein, Johannesburg 2028, South Africa

**Keywords:** cellulose, biomass, ionic liquids, carbon-based nanostructured material, energy

## Abstract

Considering its availability, renewable character and abundance in nature, this review assesses the opportunity of the application of biomass as a precursor for the production of carbon-based nanostructured materials (CNMs). CNMs are exceptionally shaped nanomaterials that possess distinctive properties, with far-reaching applicability in a number of areas, including the fabrication of sustainable and efficient energy harnessing, conversion and storage devices. This review describes CNM synthesis, properties and modification, focusing on reports using biomass as starting material. Since biomass comprises 60–90% cellulose, the current review takes into account the properties of cellulose. Noting that highly crystalline cellulose poses a difficulty in dissolution, ionic liquids (ILs) are proposed as the solvent system to dissolve the cellulose-containing biomass in generating precursors for the synthesis of CNMs. Preliminary results with cellulose and sugarcane bagasse indicate that ILs can not only be used to make the biomass available in a liquefied form as required for the floating catalyst CVD technique but also to control the heteroatom content and composition in situ for the heteroatom doping of the materials.

## 1. Introduction

Globally, there has been an increase in technological development, which aims to make life on the planet sustainable. This has resulted in an increased demand for materials being utilized in the fabrication of various products. Amongst the materials are carbon nanomaterials, first reported by Kroto and co-workers in 1985 [1], followed by Iijima in 1991 [2]. Since then, there have been many investigations and numerous reports on carbon-based nanostructured materials (CNMs). Research has been driven by various unique properties displayed by CNMs, such as the large surface area to volume ratio, high strength, enhanced optical absorption properties and superior carrier (electrons and holes) transport characteristics [3,4,5,6]. These properties render CNMs applicable in diverse functions such as catalysis [7], gas sensing [8], energy conversion [9] and storage, amongst others [10,11]. Most of these applications require a particular type of CNMs to be highly efficient. Hence, a wide range of shaped CNMs have been reported, including spheres [12], fibers [13], ribbons [14] and tubes [15].

The production of these shaped CNMs is dependent on the starting materials utilized for their synthesis, and the reaction conditions. Bai and co-workers reported that using ferrocene and benzene yields different types of CNMs, such as single-walled carbon nanotubes (SWCNTs), multiwalled carbon nanotubes (MWCNTs) and carbon nanofibers (CNFs) [16]. They concluded that the precursor to catalyst ratio determines the product. This observation has also been supported by Nyamori and Coville [17], where apart from the product distribution, the size and shape of the CNMs were also influenced by the carbon to iron (catalyst) ratio. Additionally, in another study by Sevilla et al. [18], they synthesized graphitic carbon structures using gluconate dehydrates of iron and cobalt, and reported that iron and cobalt nanoparticles successfully catalyzed the reaction at either 900 or 1000 °C. Different CNMs, i.e., worm-like and broken hollow carbon nanostructures, were synthesized using gold- and silica-containing precursors [19].

Many reports describe the production of CNMs such as carbon nanotubes (CNTs), CNFs, and carbon nanospheres (CNS) using long chain and aromatic hydrocarbons as the source of carbon in the presence of different metal salt catalysts [20,21,22,23,24]. Generally, CNMs are commonly formed through bulk pyrolysis and subsequent solvothermal treatment to purify the product. Pyrolysis involves the use of heat to decompose a mixture containing the soluble metal salt (usually the catalyst precursor) and organic compounds rich in carbon (carbon source) into char in an inert (or slightly reducing) atmosphere. In some cases, water or oxygen may be employed such that partial combustion transpires [25]. Therefore, in pyrolysis, two stages are involved; firstly, the synthesis of metal nanoparticles and, secondly, the growth of a coating material using the nanoparticles as templates [26]. On the other hand, solvothermal treatment involves the post-synthesis purification of nanomaterials using solvents in an autoclave, and the term hydrothermal treatment is used when the solvent is water. In these synthesis procedures, the major challenges faced in the production of CNMs include large scale and economical production, and overcoming low fabrication yields [18]. Therefore, there is a need for innovative research efforts that focus on employing starting materials that are cost-efficient and readily available. Hence, alternative materials to fossil-based carbon sources, such as biomass as a renewable precursor, are highly promising.

In the future, renewable resources should be used extensively, since they can be replenished relatively quickly. Biomass, i.e., the entire weight of flora and fauna in a specific region, can, in principle, be considered and explored [27]. Plant biomass includes waste residues, such as garden refuse, wood chips, corn residues, and cereal straw, just to name a few examples. The wide availability of biomass suggests a favorable alternative precursor source for CNMs. However, the complex composition of biomass and process-ability can raise the question of whether CNMs with specifically tailored properties can be designed for the intended application. In broad terms, dried biomass contains lignin (10–15%), hemicellulose (20–40%) and cellulose (40–50%). On a molecular basis, these components exhibit strong intermolecular interactions (e.g., hydrogen bonding), are polymeric and hence insoluble in most solvents [28,29]. Considering synthesis methods where liquidized feedstocks are required, this insolubility poses a major challenge. However, recently, some ionic liquids (ILs) have been identified as one of the very few solvents capable of dissolving complex biomass [30,31,32]. In particular, imidazolium- and ammonium-based ILs have been used to dissolve cellulose-containing biomass [33].

ILs are salts that have a melting point below 100 °C, below which they exist as liquids consisting entirely of cations and anions [34]. ILs are characterized by low volatility, flammability and high thermal stability. Their structural diversity can be exploited to design their physicochemical properties [35]. Their physicochemical properties make ILs not only suitable as an alternative to conventional organic solvents [36], but they also allow for the solubilization and liquid phase conversion of materials that were previously not possible. Interestingly, the application of this novel class of solvents to the conversion of biomass, and in particular to CNMs, has not been fully exploited.

As mentioned before, CNMs have outstanding properties that make them useful as part of light-harvesting devices [9]. Incorporation of CNMs in energy harvesting devices does not only makes them more robust but also enhances the light-absorbing properties of the devices, hence ultimately improving their performance [37]. Once light-absorbing properties have been enhanced, there is maximum utilization of the abundant solar energy. This is important because it facilitates the provision of reliable and secure alternative energy, contributing to the mitigation of the energy crisis. Li et al. [38] reported biomass-derived carbon as the shape controlling agent in tungsten-based nanohybrids synthesis. The nanohybrid composites resulted in enhanced electrochemical performance in dye-sensitized solar cells. For energy storage devices, i.e., supercapacitors, functionalized microporous conducting CNMs derived from biomass (Neem and Ashoka leaves) with high surface area, specific capacitance were reported [39]. This review focusses on the dissolution of cellulose-containing biomass as a precursor for the synthesis of CNMs. Additionally, the current state of the art of the application of biomass-derived CNMs in energy devices is reviewed.

## 2. Carbon-Based Nanostructured Materials

Carbon exhibits distinctive structural, physical, mechanical, electrical and chemical properties, which make it appropriate for a wide range of respective applications [40,41,42,43]. Carbon joins through different types of bonding, which results in the materialization of different allotropes, e.g., amorphous carbon, graphite, diamond, carbon nanotube and fullerenes. The resulting type of hybridization formed through the chemical bonding is responsible for the various characteristic, physical and chemical properties for each class of allotrope. Hybridization can be *sp*^2^ forming two-dimensional planar sheets resulting in graphene/graphite, or *sp*^3^ in a three-dimensional network generating a diamond structure. CNMs are categorized based on shape, composition and orientation. They can exist as spheres, fibers, tubes, onions, horns, capsules, ribbons and coils, as shown in Figure 1 [44].

Graphene sheets roll up or assemble to form various structures: sheets can form zero-dimensional, one-dimensional and three-dimensional structures, resulting in fullerenes, carbon nanotubes or graphite, respectively. The various unique nanoforms exhibited by the CNMs are responsible for their functionalities in particular applications [45]. For instance, CNMs that are porous in nature have large surface areas, which makes them suitable for several applications, including electrocatalysis [46], separation [47] and energy conversion and storage [48]. The methods of synthesis of the CNMs are discussed in the subsequent sections.

### 2.1. Synthesis of Carbon-Based Nanostructured Materials

The most commonly used methods for the synthesis of CNMs under controlled conditions include arc discharge [2,49], laser ablation [50], and chemical vapor deposition (CVD) methods [51], rather than bulk pyrolysis. The arc discharge set-up is mainly used for the decomposition of gaseous hydrocarbons, although in certain instances, mixtures with water and liquid nitrogen have also been reported [52]. Typically, the method utilizes two graphite electrodes, where direct current arc voltage is applied across, to generate some carbon atoms in between the electrodes. The reaction chamber is saturated with gas or immersed inside a liquid environment. This synthesis type uses temperatures as high as 3000 °C [53]. The arc discharge or high energy plasma is used to initiate chemical reactions; thus, a complex environment of carbon radicals, other chemical radicals, free electrons, thermal and non-thermal energy, exists between both electrodes. This step is followed by the processes of evaporation, sputtering and consumption of the anode. The anode sputtering then produces some CNMs that are partially deposited onto the cathode surface, while the remainder escapes the hot zone into the greater area of the reaction chamber [54,55]. Arc discharge synthesis in the presence of pure graphite rods results in fullerenes deposited as soot inside the chamber, and MWCNTs are deposited on the cathode. If synthesis employs a graphite anode comprising of a metal catalyst, and a pure graphite cathode, SWCNTs are produced in the form of soot. In the case of laser ablation, a pulsed or continuous high power laser beam is used. The high energy from the laser beam is responsible for the ablation of the starting materials (either solid or liquid). Usually, the precursors consist of a solid mixture of graphite and a catalyst, and in the presence of an inert atmosphere, they are vaporized at temperatures greater than 3000 °C [56]. When the target is vaporized, a carbon-based plume collects on a cooled part on the inside of the apparatus. The CVD method involves a reaction chamber in which a substrate is exposed to the starting materials. Ideally, the starting material will be volatile such that it can be transported by a carrier gas and then reacts, due to the applied temperature within the reactor, to form CNMs on the substrate. In this case, the synthesis temperature is relatively lower and can be in the range of 200–1100 °C.

In terms of drawbacks, arc discharge and laser ablation methods are associated with various challenges, including high energy usage, tangling of products, poor product purification and small scale of production, which renders the CVD method the most suitable type of synthesis [51]. In the CVD technique, a volatile carbon source, in the presence or absence of a catalyst, is converted into a non-volatile solid product via the gas phase. Advantageously, gaseous or liquid precursors can be injected directly into the chamber, i.e., by the floating catalyst CVD method [22,57,58]. This synthesis technique is associated with several advantages in the production of CNMs. These include high productivity, excellent quality, controlled growth and good uniformity and relative ease of controlled doping of CNMs with heteroatoms [59,60,61]. In the synthesis of carbon spheres and fullerenes, the CVD technique has the major advantage of producing uniform sized structures. In the case of carbon nanofibers synthesis, the CVD technique is advantageous in tailoring the diameter, crystallinity, and also orientation of the fiber axis [59,62]. In CNTs production, the CVD technique has several advantages, including exceptional control over the carbon to catalyst ratio and the product dimensions (length and diameter), as well as alignment [59]. Due to the solid nature of most biomass, the direct application of the floating catalyst CVD technique is little explored. Irrespective of the synthesis method, only a few reports exist that describe the conversion of biomass to CNMs, and these include rice husks [63], grape seeds [64], wood [65], and corncobs [66], among other examples [67]. However, the limitation in the aforementioned studies is that previously carbonized biomass is employed as a precursor in its solid form resulting in little control on the shape and size of the nanomaterials produced. The main challenge is the direct conversion of cellulose-containing biomass materials into the CNM of choice.

To successfully synthesize the CNMs, several reaction parameters, such as temperature, atmosphere, time, catalyst and carbon source, play a significant role and can be controlled [68,69].

The inclusion of a catalyst source in the reaction mixture or reaction chamber will result in a unique interaction between the organic precursors and the catalyst involved. The catalyst, when in its active phase, is capable of modifying the type of carbon radicals that are formed during the decomposition of the carbon source [70]. These carbon radicals are crucial in the growth of CNMs, and the catalyst acts as a nucleation site for the formation of CNMs [71,72,73]. Thus, the catalyst facilitates the graphitic structure formation uniquely, depending on the type of association between the catalyst and the carbon radical. Recently, various authors have reviewed the importance of the catalyst for the controlled growth and induction of chirality of single-walled nanotubes [71], and the selective synthesis of various multiwalled nanotubes [74], graphene [75], carbon dots [76], and other types of carbon nanomaterials [77]. For example, Bai et al. studied the influence of both the catalyst and carbon source, ferrocene and benzene, respectively, using the floating catalyst CVD method on the production of CNTs and CNFs [16]. The diameter and the structure of the products were dependent on controlling the ratio of the precursor to the catalyst. The diameter of the products was reduced upon the increase in the precursor to catalyst ratio whilst the products also became more twisted. Nyamori and Coville found that both the carbon to catalyst ratio and the temperature affected the shape of the resulting CNMs [17].

Organometallic catalysts [78], metal oxides, and single or bimetallic transition metal catalysts [79] have been used to synthesize CNMs. Of particular importance is the utilization of transition metals in the conversion of biomass [80]. Transition metals such as Fe, Co and Ni possess higher melting points along with lower equilibrium vapor pressures when compared to main group metals, which qualifies them as suitable catalysts for numerous hydrocarbon species during synthesis. Furthermore, Fe, Co and Ni catalysts exhibit a high solubility of carbon, superior carbon diffusion rates, and they form meta-stable carbides [81,82]. These properties are responsible for the growth of the crystalline tubular structures [83]. Considering the specific role of the catalyst in the conversion of biomass to CNMs, Collard et al. evaluated the influence of Ni and Fe salts on pyrolysis [84], suggesting that yields obtained from beech wood depend on the metal catalyst employed. A higher char yield and reduced tar formation were observed with iron-impregnated wood compared to nickel-containing precursors.

Other studies have utilized ILs as carbon precursors for the preparation of CNMs. In particular, Dai and co-workers prepared CNMs by carbonizing nitrile-functionalized IL precursors [85]. Another study explored ILs for heteroatom doping of CNMs. For this purpose, sodium glutamate, tartrate, or citrate were doped with ILs, i.e., 1-butyl-3-methylimidazolium chlorine and 3-butyl-4-methythiazolium bromide) [86]. All these ILs demonstrate the possibility of CNMs synthesis in the absence of an organic solvent. A summary of the processes involved in the fabrication of CNMs is provided in Scheme 1.

### 2.2. Biomass as a Renewable Carbon Source for CNM Synthesis

As aforementioned, the literature on the application of native biomass to CNM synthesis is relatively limited. This may partially be due to the fact that biomass conversion is still in its infancy in general, and partially because biomass, with its inherent complex composition and properties, may require adaption of protocols generally used. For example, native biomass is inherently insoluble in most solvents, rendering feedstock dosage difficult. Of course, one may opt for the application of biomass-based chemicals rather than native biomass [87,88,89]. However, for this review, the focus is placed on direct biomass conversion to CNMs.

It has been stated that polymeric materials, which upon heating undergo solid–gas transitions without melting, are considered good potential precursors for CNMs [90]. Interestingly, lignocellulosic biomass is composed of different polymers, including cellulose, hemicellulose, and lignin. Large quantities of lignocellulose are produced as residue in agro-processing and forestry activities, which are of little nutritional value, and therefore represent a valuable chemical and material resource. Dried biomass mainly comprises hemicellulose (20–40%) cellulose (40–50%), lignin, extractives, as well as inorganics [91]. Cellulose is the structural component, a fundamental reinforcement unit of the primary cell wall of green plants, and can be produced by a large variety of living organisms. It is the most abundant organic compound which has outstanding properties, including biocompatibility, biodegradability, and relatively high thermal and chemical stability [92]. Cellulose is a linear molecule with a high molecular weight and a partially crystalline structure [93]. Interestingly, it can be expected that cellulose, a polymeric material, when exposed to extreme heat, is indeed capable of producing carbon residue without melting, thus fulfills one of the requirements of suitable CNM starting materials [90]. During pyrolysis, gaseous products such as CO_2_, CO, alcohols, ketones and other low molecular weight carbon-containing substances are produced from biomass.

On the other hand, previous work indicated that a low oxygen content of the starting material is advantageous in CNM synthesis [68], as typically found in fossil-based hydrocarbon precursors, including toluene, methane, acetylene and benzene. This may hamper the application of biomass as a precursor, due to its inherently high oxygen content, as found in particular in carbohydrates.

From the scarce literature on CNM synthesis from biomass, it can be derived that these aspects may be responsible for low yields in pyrolysis, in particular since some biomass-based processes require a hydrothermal pretreatment or other activating processes [94,95]. For example, Gan et al. describe the hydrothermal pretreatment of sodium carboxymethylcellulose in an autoclave, followed by extraction and filtration, before pyrolysis of the resulting carbonaceous solid at 800 °C under nitrogen [96]. Similarly, sodium alginate [97], cellulose [98] or sugarcane bagasse [99] were submitted to carbonization under a nitrogen atmosphere and chemical activation [97] prior to pyrolysis.

Evidently, any pretreatment step increases energy, solvent and chemical expenditure, and should be avoided. Indeed, Dubrovina et al. reported the direct bulk pyrolysis to yield CNTs, starting from cellulose acetate cross-linked with polyisocyanate using NiCl_2_ as the pre-catalyst [100]. Although this demonstrated the feasibility of producing CNTs directly from bio-based starting materials, bulk pyrolysis resulted in low yields and selectivities.

Hence, the ability to use higher-yielding and more selective methods, such as the floating catalyst CVD technique, would be highly beneficial but is hindered by the often solid nature of biomass. It was hypothesized that this aspect could be overcome by utilizing solvents with an ability to dissolve biomass. Biomass features strong inter- and intramolecular interactions through hydrogen bonding, rendering it insoluble in aqueous and most organic solvents.

#### 2.2.1. Biomass Dissolution

Driven by the quest to make spinnable fibres (rayon, viscose), early research focused on the dissolution of cellulose rather than native biomass. The dissolution of cellulose was achieved by derivatization, e.g., in the cuprammonium and xanthate processes [101], which were industrially implemented already at the end of the 19th century. These processes require the use of high ionic strength solvents/reagents, relatively harsh conditions, and are expensive and cumbersome [102,103]. The chief disadvantage associated with these processes is, however, related to emissions to the environment since the reagents and solvents can often neither be recovered nor be reused. However, in 1934, Graenacher discovered that the physical dissolution of cellulose is possible in molten salts such as *N*-ethylpyridinium chloride in the presence of a nitrogen-containing base [104]. Unfortunately, these salts possess relatively high melting points and result in cellulose solutions with high viscosity, which are difficult to process. Further research showed that various other solvent systems such as *N*-methylmorpholine-*N*-oxide [105], *N*,*N*-dimethylacetamide/lithium chloride [106], 1,3-dimethyl-2-imidazolidinone/lithium chloride [107] and dimethyl sulfoxide/tetrabutylammonium fluoride [108] are capable of dissolving cellulose. Although some of them were industrially implemented, none fully satisfy the requirements for the sustainable production of fibers [109,110]. Therefore, the challenge to design ‘greener’ alternative solvents continued.

#### 2.2.2. Ionic Liquids in Dissolution of Biomass

Possibly the earliest report of a room temperature IL was produced when Walden investigated the effect of ion size on conductivity, in 1914, and found that *N*-ethylpyridinium nitrate is a low melting salt [111]. However, the implications of salts melting at or below room temperature were not realized (with few exceptions) in the decades thereafter, until the mid-1980s. ILs are salts with low melting temperatures (below the boiling point of water), which are made up of anions and cations [112]. These ILs have negligible vapor pressure, high thermal stability, high ionic conductivity and ability to solvate compounds of widely varying polarity [113,114,115]. In addition, ILs are structurally highly flexible; they allow for chemical tuning of their physicochemical properties according to requirement by a suitable choice of cations and anions [116]. Tuneable properties include not only the physical characteristics such as melting point, viscosity, solubility, density, but also the chemical characteristics, i.e., hydrophobicity, acidity/basicity and specific interactions with solutes. Therefore, all the properties associated with ILs make them suitable for many synthetic and catalytic processes [117,118,119,120], and in particular, for those where conventional organic solvents fail to provide solubility.

The early work of Swatloski et al. [121] pioneered a large body of work into the dissolution of cellulose in ionic liquids. For highly hydrogen-bonded molecules occurring in biomass, a careful choice of anion and cation of the IL is crucial. In general, if the cation head group and the anion is kept constant, and the alkyl substituent lengthened, the solubility decreases. Whether this is due to specific interactions, or rather the increasing viscosity of the ionic liquids, is still being debated. In addition to effects associated with both the cation and the anion size and steric aspects, [122] an antagonistic interaction between the cation and anion of the IL with hydroxyl groups of cellulose was proposed (Figure 2), based on molecular dynamics simulations [123].

The hydrogen bond acceptor strength [125] of the anion appears to be the dominating aspect, while the cellulose–cation interaction may occur either via hydrogen bonding or charge delocalization [126,127,128,129,130]. The cellulose solubility in ILs can be further increased by either including electron-withdrawing groups in the alkyl chains of the cation, while electron-withdrawing groups have a detrimental effect. A summary of the cellulose solubility with regards to different cation–anion combinations is provided in Figure 3. Small cationic chained (C1-C4) ILs possess a greater cellulose solubility, followed with acetate containing imidazolium-based ILs. The high solubility has been associated with the hydrogen bonding basicity, less steric hindrance and the ability to break inter- and intramolecular hydrogen bonds [33].

### 2.3. Biomass-Derived Carbon-Based Nanostructured Materials

#### 2.3.1. Graphene

Structurally, graphene exists as a single atomic layer of graphite and is also the basic structural unit for fullerenes and CNTs [131]. Graphene is made up of *sp*^2^ and *sp*^3^ hybridized carbon atoms closely packed into a honeycomb lattice by reference to δ bonding in which the presence of delocalized electrons is due to π orbitals of carbon atoms [132]. Graphene is a two-dimensional material that possesses exceptional electrical, mechanical and thermal properties, thus, making it suitable in energy-related devices, amongst other applications [133].

Traditionally, graphene is produced from single chemical precursors [134]. However, the use of more complex biomass-derived materials has been described in a few publications. For example, chitosan produced from crustacean waste yielded a single layer of nitrogen-doped graphene, when pyrolyzed at 800 °C [135]. Shams et al. reported on the use of camphor leaf to produce graphene using pyrolysis at 1200 °C [136], followed by an elaborate purification step to separate graphene from amorphous carbon. Chen et al. successfully synthesized graphene from wheat straw, using a combined hydrothermal and graphitization approach [137]. Interestingly, when graphene oxide and milled sheep horn was pyrolyzed, a three-dimensional porous, nitrogen- and sulfur-doped graphene was found, demonstrating the potential of using biomass as both the carbon precursor and the source of functionalization [138]. In neither case did the synthesis require catalysts nor strong acids, making the production of graphene from renewable materials an attractive area of research with interesting prospects.

#### 2.3.2. Carbon Nanotubes (CNTs)

CNTs are extremely lightweight seamless hollow cylinders formed by covalently bonded *sp*^2^ carbon atoms. The hybridization type is responsible for their mechanical properties exhibited by these structures, including high mechanical and tensile strength, toughness and stiffness, making them suitable as materials for reinforcement applications [139]. Additionally, CNTs have other unique properties such as high electrical conductivity [140], high thermal conductivity [141], flexibility [142], and attractive optical properties [143]. When a single graphene sheet is rolled up at specific sites to form a seamless cylinder, this results in patterns that determine whether the material exhibits semiconducting or metallic properties. Metallic or semiconducting properties are usually only observed with single or double-walled CNTs. The metallic type is associated with very high conductivity, which is higher than that of copper. CNTs can exhibit high thermal conductivity and heat capacity, and these aspects have been exploited in composites or systems for heat dissipation [144,145]. The optical properties of CNTs vary from near-ideal black body absorber to transparent, within composites, and as such, CNTs have been applied in electromagnetic interference shielding, transparent coatings and films [145,146].

A few examples of the synthesis of CNTs from both cellulose and lignocellulosic biomass have been reported, making use of bulk pyrolysis [147]. For example, absorbent cotton [148], after treatment at temperatures of 400–600 °C, yielded CNTs with inner and outer diameters of 10 and 80 nm, respectively, after a purification step involving ethanol and water. Cellulose acetate [149], pyrolyzed in the presence of polyisocyanate and the pre-catalyst NiCl_2_ at 750 °C, yielded CNTs with 24–38 nm diameter.

CNTs were also obtained from lignocellulosic biomass, such as wood sawdust [150] or wood fiber [87]. In the latter case, diameters ranged between 10 and 20 nm when the carbonization was carried out at temperatures between 240 and 400 °C under oxidative conditions. Purification included an extraction with HCl [87]. The microwave-induced pyrolysis of gumwood [151] at 500 °C yielded CNTs with 50 nm diameter in the presence of SiC and nitrogen as an inert atmosphere. Pyrolysis-derived bamboo charcoal in the presence of ethanol vapor was used to produce CNTs at high temperatures of 1000–1500 °C using CVD [152]. MWCNTs with diameters between 30 and 50 nm were produced from the grass at 600 °C [153].

These examples demonstrate that CNTs can be successfully produced from biomass. However, little is currently known on how to control diameters or lengths of the CNTs hence produced.

#### 2.3.3. Carbon Nanofibers (CNFs)

Another type of CNMs is carbon nanofibers (CNFs), which are *sp*^2^-based nanostructured materials that are straight. They resemble CNTs under a low-resolution scanning electron microscope (SEM). However, under a transmission electron microscope (TEM), they appear as a solid material, as they lack the hollow tubular structure exhibited by CNTs. CNFs have exceptional mechanical properties that enable them to be applied as reinforcement material in composites. Additionally, CNFs have outstanding electrical conductivity, which makes them suitable for energy and self-sensing device applications.

The production of CNFs has been reported from bacterial cellulose, which had been pre-treated with an aqueous solution containing doping heteroatoms(s), prior to pyrolysis [154,155], resulting in three-dimensional heteroatom-doped (P or N/P or B/P) CNFs. The extent of nitrogen-doping was adjustable by controlling the concentration of ammonia in the solution [156]. In these studies, the resulting material has to be purified by extraction with acid, because of the synthesis methodology followed. It is difficult to tailor material properties due to acid treatment because acid treatment induces some surface and morphological changes.

#### 2.3.4. Carbon Onions

Carbon onions are *sp*^2^ bonded multi-layered quasi-spherical and polyhedral shaped shells with concentrically layered structures [45]. This type of material possesses different chemical and physical properties, which include high-quality optical properties and moderate capacitance; hence, they are applied in supercapacitors, as both active materials and easily dispersible conductive additives [45,157]. Only one report exists where carbon onions were synthesized from biomass, namely by pyrolysis of wood wool [158] at 600 °C in an atmosphere comprising of a mixture of nitrogen and oxygen.

#### 2.3.5. Carbon Spheres

Carbon spheres (CSs) are layers of carbon that are a result of broken concentric layers emanating from the core. They are held together by van der Waals forces in agglomerates [72]. CSs have been used as part of the electrode material in supercapacitors [159] and lithium-ion batteries [160]. From biomass, CSs were produced from either carrageenan or cassava/tapioca flour as the precursors utilizing the hydrothermal carbonization method with some chemical activation [161,162].

## 3. Surface Modifications of CNMs

Synthesized CNMs agglomerate due to strong van der Waals interactions [78]; thus, it is necessary to disperse after synthesis. During the purification process, dispersion is achievable. In some instances, the process results in the introduction of some functional groups on the surface of the CNMs, thus modifying the as-synthesized structures. The main approaches involved in the modification of CNMs fall into three major categories. These include covalent bonding of chemical groups onto the π-conjugated skeleton of CNMs, non-covalent adsorption or covering with several functional molecules and endohedral filling of their empty inner cavity [163]. The covalent modification of CNMs can be of two types, i.e., on the sides of the CNMs structure or at defect sites, dependent on the location of the functional groups [164]. In fullerenes, functionalization has been successful with both electron-donating and withdrawing groups on aromatic rings [165]. Surface modifications in fullerenes differ from the aromatic system reaction patterns. Reactions during fullerene surface modification resemble reactions of localized electron-deficient polyolefins, i.e., electrophilic, dienophilic and dipolarophilic. The fullerenes undergo cycloaddition reactions, react with several nucleophiles (amines, phosphines, Grignard and organolithium reagents), and undergo radical addition reactions and transition metal complex formations resulting in surface modifications [166]. Considering graphene-based CNMs, these are difficult to functionalize due to the high interlayer cohesive energy [164]. Therefore, specific chemical reactions are necessary for modification, e.g., through click chemistry, i.e., cycloaddition, biomolecules functionalization, fluorination/bromination, electrophilic addition, radical addition, nucleophilic addition, and grafting of polymers. These reactions are well reported by Karfa and co-workers [164]. In general, electrical and mechanical properties of covalently modified CNMs change due to structural defects initiated by carbon–carbon double bond breakages on their graphitic matrix during associated chemical processes. Figure 4 shows examples of the chemical modification of the surface with various functional groups.

Non-covalent approaches through either the adsorption (for example, surfactants or bio-macromolecules) or π–π stacking of small aromatic molecules and conjugated polymers can help preserve the intrinsic properties of CNMs, but may result in leaching when used in solution [167]. In the case of water treatment, CNMs are modified with functional groups that provide interaction with undesirable metal ions. In energy storage and conversion devices, CNMs are modified with various chemical groups, including polymers. The modification also results in the solubility of the CNMs in respective solutions.

## 4. Application of Biomass-Derived Carbon-Based Nanostructured Materials in Energy-Related Devices

The modern world is experiencing an ever-increasing demand for energy. In particular, in developing countries, driven by population growth and industrial advancements, power cuts are experienced frequently due to the inconsistent availability of suitable fossil fuels, as well as centralized power generation units providing electricity into an underdeveloped network that is infrequently upgraded and maintained [168]. Insufficient provision of electricity hugely affects economic growth and national security. Therefore, harnessing renewable energy sources (in particular hydro, wind, solar and tidal power) to provide electricity in a decentralized fashion remains the top priority for any growing economy [169,170,171]. The capture, conversion and storage of solar energy have been extensively investigated [9,172,173,174], as it is abundantly available in many developing countries.

### 4.1. Energy Conversion: Solar Cells

Solar energy can be harnessed and converted to electrical energy either through thermal conversion or by photovoltaic (PV) direct conversion into electricity through the use of solar cells. The PV conversion process involves rapid charge separation, transport and collection; thus, these factors affect the efficiency of solar cells. Studies have reported that the incorporation of CNMs as part of the solar cell structures has resulted in improved efficiency and overall device properties [37].

There are several types of solar cells available to date, and these include silicon-based [175], and other inorganic-based [176], hybrid [177] and organic solar cells [174]. Due to their high power conversion efficiencies (up to 25%), silicon-based solar cells are the preferred inorganic solar cells [178] although their wide-spread implementation is hampered by the high energy requirement and associated cost of their production [179]. This aspect was improved on by inorganic thin-film technology, also resulting, however, in reduced efficiencies due to lack of crystallinity of the semiconducting layer. CNMs-based solar cell efficiencies are represented in Figure 5.

#### 4.1.1. Organic Solar Cells

As an alternative to inorganic solar cells, organic solar cells (OSC) were developed over the last 30 years. They feature several advantages, including structural flexibility, extraordinary optical absorption coefficients, and the ability to produce very thin solar cells. The semiconducting material in OSCs possesses both the ability to absorb light in the UV-Vis range of the solar spectrum and to transport electric current. For example, in the bulk heterojunction organic solar cell (BHJ-OSC), an interpenetrating network of an electron donor, usually the polymer, and an electron acceptor, normally a fullerene derivative, is created. The role of the polymer is to absorb light and, thus, create electron-hole pairs. Once these excitons are generated, they disperse in the direction of the donor–acceptor interface, where the electron-donor-acceptor complexes are formed [181]. The electron–donor–acceptor complexes are susceptible to either generation or recombination, which quantifies the available free charge carriers. The electron and electron-hole of the free charge carriers are then moved by an internal electric field and extracted by particular electrodes [182].

Movla et al. investigated the performance of a BHJ-OSC-based on poly(3-hexylthiophene) (P3HT) and phenyl-C61-butyric acid methyl ester (PCBM), P3HT:PCBM by solving the drift-diffusion equations [183]. In their model, they took into account the boundary condition and uniform potential energy, the effects of the active layer thickness and the photocurrent generation. The data obtained were for both in the dark and under illumination. They concluded that the device performance relied on the thickness of the active layer, the materials and methodology used for fabricating the devices.

The demerits allied with OSCs include low power conversion efficiency, which for tandem solar cells is currently around 12.5% [184], low stability (moisture) and strength. In order to address these challenges, the engineering of new materials and device structures is of importance. Efforts to design and investigate innovative methods to convert solar radiance into electricity have seen the development of hybrid solar cells (HSC).

#### 4.1.2. Hybrid Solar Cells

Hybrid solar cells (HSCs) are constructed with a combination of nanostructured organic and inorganic semiconducting materials to make use of their respective advantages. HSCs have the advantage over OSC that they possess high carrier mobility, and the onset of absorption is at shorter wavelengths [185]. Additionally, HSCs are capable of being processed with ease, allowing for the manufacture of printable devices in a reel to reel fashion at a greater speed and reduced cost [185]. Ideally, an *n*-type inorganic composite material is coupled to a *p*-type conjugated polymer or otherwise an *n*-type organic material and *p*-type inorganic semiconductor [186]. In an HSC, organic materials, usually conjugated polymers, are the donor material responsible for the transportation of holes. On the other hand, the inorganic semiconductor, which can be constructed from nanoparticles, works as acceptors accountable for the movement of electrons. Inorganic material is allied with environmental stability, elevated carrier mobility, well-suited fabricating processes, high absorption coefficients and the ability to be produced in different sizes [187]. Nanoparticle size and shape modification, when considering quantum confinement, influence the confining dimensions hence adjusting the bandgap, which in turn alters the absorption profile [188]. Collectively, the organic and inorganic hybrid systems have unique and desirable properties ideal for distinct solar cell devices [189,190,191].

Generally, hybrid solar cells are planar. In this manner, they consist of an anode, substrate, photoactive layer and a cathode, each with a different function from the other. The organizational structure is in such a way that a flexible and transparent material is used as a substrate; this may be glass or plastic. An anode is deposited on top of the substrate, e.g., an oxide layer possessing semi-transparent properties. The purpose of the anode is to permit light to go through and thus accumulate the holes from the device. A conductive mixture is employed on top of the anode. This mixture smoothens the surface of the anode, prevents oxygen from penetrating the active layer and saves as the hole transporting layer and exciton blocker. The photoactive material is then deposited on top of the conductive mixture. Finally, the cathode, Al, Ca or Mg, is deposited on top such that it collects electrons from the device. Generally, the photoactive material is hosted in between electrodes of different work functions [192].

The operating principle involves light absorption, exciton generation and diffusion, exciton dissociation to carriers at the junction interface and carrier transportation and collection. Typically, photons that possess energy greater than the bandgap of the organic or inorganic semiconductors, incident photons, are absorbed, and this results in excitation of electrons in their ground states. This generates excitons that are bound to electron-hole pairs. This, in turn, results in exciton diffusion to respective interfaces through organic/inorganic semiconductors. Because the organic–inorganic hybrid p-n junction comprises of an inner electrical field, built excitons are dissociated into holes and electrons at the interface. Charge carriers are transported to respective electrodes to generate an external current. Finally, holes are transported to the anode electrode through donor materials, electrons to the cathode through the acceptor materials [193].

Figure 6 shows the summary of events that occur, i.e., electrons are created in the conjugate polymer while holes are created in the inorganic nanoparticles. A favorable situation is when the HOMO and LUMO energies of the conducting polymers are at a higher level than the valence band and the conduction band of the semiconductor.

Light absorption is optimal using conducting transparent films. Transparency and conductivity thus complement each other, and together with other properties like low infrared emittance, the overall result is a good electrical conductor. A good electrical conduction will result in the application of films in different electrical devices [173]. Indium tin oxide and fluorine tin oxide materials are typical examples of transparent films that have been reliably used; however, there has been some research replacing these with relatively cheaper materials that have competitive properties such as graphene and other CNMs [194]. The advantages and disadvantages of the use of CNMs in OSCs have been reviewed [195]. Due to their conducting nature, CNMs are incorporated in the active layer of the OSCs. In polymeric materials, CNMs combine with the π-electrons of conjugated polymers and act as electron acceptors. In addition, these CNMs improve the dissociation of excitons through an enhanced electric field at the interface between the CNM and the polymer, thus restraining the recombination of photogenerated charges. The addition of CNMs in OSCs enhances charge separation and improves the transfer of charge carriers to the electrodes before they recombine. For example, a single sheet of graphene possesses ~2.5% of absorption [196]; therefore, incorporating graphene as part of the active layer contributes significantly to the device performance. Considering the excellent thermal properties of CNMs, their incorporation into OSCs can reduce the photodegradation of OSCs. Enhanced performance of OSCs with CNMs has been attributed to either decreased carrier recombination, improved electrical conductivity in the active layer, or enhanced dissociation of excitons.

Despite these positive contributions in OSCs, the power conversion efficiency of CNMs-based cells is still low compared with that of tandem solar cells. There are several limiting factors associated with CNMs; these include the incomplete exciton dissociation, particularly for the small quantities of CNMs at the percolative threshold. On the other hand, increasing the CNMs ratio in the polymer matrix to rectify this challenge intensifies shunts and recombination pathways as a result of their random distribution and high aspect ratio. Therefore, these disadvantages suggest that a major strategy for the maximization of free charge carriers generation and power conversion efficiency could be the precise separation of CNMs. The charge mobility enhancement in OSC has been reported with the use of boron and nitrogen-doping in CNMs. Doping CNMs with boron or nitrogen selectively enhances charge mobility and dispersibility in the polymer matrix and also tunes the work function [197]. In general, the limited work on the use of CNMs in OSCs is due to difficulties in processing CNMs, producing a highly uniform and well-characterized sample. Hence, very little or no work on CNMs from biomass and their use in OSCs is available. Therefore, using ILs to produce a liquid feedstock, which can be used to produce CNMs with tailorable characteristics, such as desired dimensions, doped with heteroatoms such as B and N, will open new pathways to use rationally designed CNMs for OSCs.

### 4.2. Energy Storage

Batteries and electrochemical capacitors are currently prominent energy storage devices. Electrochemical capacitors are characterized by fast charging-discharging rates and long cycle life. Batteries such as Li-S, are characterized by low capacities, and low Coulombic efficiencies with poor cycle stability due to the presence of S. However, the main drawback of this type of device is that it has a low energy density [198]. Relatively little work has been conducted on the incorporation of biomass-derived CNMs into these energy storage devices to enhance performance. Hence, there is also a need to design and control the synthesis of CNMs to suit the desired application.

#### 4.2.1. Batteries

Studies have used biomass material to produce hierarchical porous carbons to enhance performance in batteries [136,199,200,201]. Cattle bone was used to fabricate Tremella-like hierarchically porous carbon nanosheets [202]. The resulting carbon material had great surface area and high heteroatom content, such that upon application in batteries, outstanding rate performance and long-term cycle stability was reported. In another study, coconut oil was used as a carbon source to produce carbon nanoparticles [203]. The carbon nanoparticles were used as anode materials in both Na and Li-ion batteries, were they showed improved storage capacity as well as stable cycling. In Li-S batteries, the cathode material was prepared from soybean hulls, and a good electrochemical performance was reported [204]. Another S cathode stabilizing agent was made from mandarin peels, the carbon contributed to excellent reversibility and cycling stability of the Li-S battery [205].

#### 4.2.2. Supercapacitors

Owing to alterable physicochemical properties, CNMs have been incorporated within energy storage systems in particular as electrode material [206,207]. The development of an electrode permitting a high-end specific capacitance is meticulously associated with the availability of interlocked meso- and micro-pores with high rate capability. CNMs with well interlocked minute and sizable pores would afford the opportunity to optimize the specific capacitance and rate capability of CNMs as supercapacitor electrodes. Ideally, a material with a narrow pore size distribution, where the pore size is not too small but also not too large (surface area vs. pore volume) is required. Porous CNMs, when used as electrodes in electrochemical capacitors, experience electrode kinetic challenges attributed to the inner-pore ion transport, thus causing poor performance [208]. Mesoporous activated carbon material with a significantly larger surface area was produced from bamboo and suggested to be applicable in supercapacitors [209]. The synthesized materials had an optimal mesoporous structure with a high specific surface area of 2221.1 m^2^·g^−1^, the highest capacitance of 293 F·g^−1^ at 0.5 A·g^−1^ in 3 M KOH aqueous electrolyte and an excellent rate capability of 193.8 F·g^−1^ at 20 A·g^−1^. In another study, coconut shell has been utilized for the fabrication of porous graphene-like nanosheets [210]. Theses nanosheets were used as electrodes for supercapacitors, where they displayed good cycle durability and coulombic efficiency with outstanding capacitance. These properties were attributed to the large surface area, good electrical conductivity and large pore volume of the material. Alternanthera philoxeroides, an aquatic plant, has also been investigated in the production of activated carbon for use as electrode material in supercapacitors [211]. Dandelion pappus- and wood-based nanocellulose fibrils were blended to form films [212]. These films were then pyrolyzed using low-pressure conditions and a carbon monoxide atmosphere. The synthesized material was utilized as a supercapacitor electrode material, and good specific capacitance with corresponding surface resistance was reported. A study reported by Ruan et al. [213] utilized a one-pot reductive amination process to functionalize cellulose beads with chitosan and l-cysteine. They fabricated single N- and dual N/S-doped materials. These materials, when used as electrode material in supercapacitors, displayed a specific capacitance of up to 242 F·g^−1^ and excellent cycle stability. In most studies, biomass-derived porous carbon materials were reported to be suitable for application as electrode material in supercapacitors [214,215,216,217,218,219,220,221] with enhanced performance over non-biomass material [222]. Most of these methods used a direct pyrolysis approach, or carbonization of the biomass, which seems to offer limited control on the physical-chemical properties of the CNMs. The use of ILs to dissolve biomass provides a route to produce a precursor that can be used to exploit the relatively well-developed methods for CVD production of CNMs with well-designed and desirable properties.

## 5. Summary and Key Findings

For the synthesis of CNMs under controlled conditions, CVD has shown great promise for the production of untangled products with high product purity at a relatively lower energy usage and larger production scale than arc discharge or laser ablation. In particular, the floating catalyst CVD method features high productivity, controlled growth and good uniformity for a multitude of CNM morphologies. The applicability of various CNMs to solar energy conversion and storage has been demonstrated, and it can be expected that their unique physical and electrical properties may be exploited to further increase the devices’ efficiency. Efforts to harness solar energy as one of the major energy sources for humankind in the near future must be accompanied by material development from resources that are likewise renewable. Ionic liquids, with their unique tuneability of the solubility of biomass, may play a pivotal role in the future. Ionic liquids can be used not only to make the biomass available in a liquefied form as required for the floating catalyst CVD technique, but also to control the heteroatom content and composition in situ for the heteroatom doping of the materials. The proof of concept was recently delivered for both cellulose [223] and bagasse [224]. This work hence opens avenues for future cooperative research in material development and applications.

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
