# Peer review of "Synthesis of Carbon Nanomaterials from Biomass Utilizing Ionic Liquids for Potential Application in Solar Energy Conversion and Storage"

_materials, 2020, doi:10.3390/ma13183945_

Round 1

Reviewer 1 Report

The authors describe the use of biomass for the production of carbon nanostructures. IN the introduction, authors should carefully control their manuscript, sometimes their parts seem to be cut/pasted, appear unlinked while a more integrated text where elements are consequently and logically organized to provide a complete information.

The title of the manuscript has to be changed since the focus is on ionic liquids. Apart from the section 2.2.2 where the dissolution of cellulose is described, the generation of carbon nanostgructures described in the next sections is essentially based on pyrolysis. Also energy conversion is to general since many different forms of energy might be considered. Solar energy could be used.

Row 62 the phrase “Mechanistically…” appears to be out of context. Please develop the topic or drop the phrase.

Row 131 what is a liquid atmosphere? Liquid environment maybe? In addition more information regarding the arc discharge process should be given. The reader should obtain information how a discharge leads to formation of CMSs.

Same holds for laser ablation which, by the way, is performed also in liquids.

Row 146 CVD synthesis of carbon nanostructures not only carbon nanotubes should be addressed (see graphene for example). In a review article the authors should introduce the topic in a general form and then report with examples. Here the single case, for instance the synthesis of carbon nanotubes, is used as to describe the overall properties of the CVD technique.

Row 301: graphene as a structural unit for fullerenes. This is a very uncommon process needing a reference.

Section 3 funcitonalization of carbon nanomaterials. This section is dedicated to the description of the CNT functionalization. This section has to be expanded to cover the functionalization of the other carbon nanostructures.

Scheme 2.1. The scheme does not reflects what is written in the sections dedicated to the CNMs. In particular ionic liquid dissolution, hydrothermal carbonization and pyrolysis appear as a step of the CNMs synthesis needed to produce the precursors for CVD or arc discharge. This is not mentioned in the previous sections which must be corrected considering this point or the scheme ahs to be changed.

Row 505 CNMs and conductivity. It is well known that CNMS are utilized to increase conductivity in polymeric materials. However CNMs are mainly sp2 with high absorption coefficients. Although a low concentration of CNMs is utilized, light absorption is non negligible. For example the single sheet of graphene possess ~2.5% of absorption. This point should be clarified.

Reviewer 2 Report

Please see the attached document

Round 2

Reviewer 1 Report

//

Reviewer 2 Report

The manuscript is well revised by the authors. Could be accepted to publish.